# Structure-Guided Framework for Characterizing Drug Resistance-Mediating ABC Transporters in *Coccidioides immitis*

Avelyn Jing

*The Harker School*

*500 Saratoga Avenue, San Jose, CA 95129*

United States of America

avelynjing29@gmail.com

*Abstract*—**The increasing incidence of drug resistance and the spread of fungal diseases underscore the urgent need to investigate resistance mechanisms in Valley fever, a fungal infection caused by *Coccidioides* spp. that has increased sharply in recent years and mirrors broader antifungal resistance trends. ATP-binding cassette (ABC) transporters, which are shown to efflux drugs in well-studied fungi, remain structurally uncharacterized in *Coccidioides immitis*. This study proposes the first structure-guided framework for systematic binding pocket assessment and inhibitor testing across five *C. immitis* ABC transporters. High-confidence protein structures (mean pLDDT $>$ 95) were predicted using AlphaFold2, and predicted pockets were identified using PrankWeb. AutoDock Vina docked five chemically diverse ligands, generating 520 protein-ligand-pocket complexes. Static filtering (based on docking scores and pocket probability) and reference protein alignment created a shortlist of 26 complexes, which underwent short explicit-solvent MD simulations to assess binding persistence. Ligand center-of-mass drift was used to evaluate binding retention, and 17 pockets with minimal drift were given the initial classification as stable within the 2 ns MD window. Extended 20 ns simulations on a representative subset confirmed that early-screened stable pockets generally persisted, validating the use of 2 ns MD as a pocket prioritization strategy. This work provides the first structural dynamics dataset for *C. immitis* ABC transporters, identifies promising binding pockets, and highlights milbemycin oxime as a consistent binder. The presented framework enables early-stage screening and filtering for pocket prioritization in fungal resistance–mediating ABC transporters, supporting precision antifungal development through a structure-guided analysis of transporter pockets.**

*Index Terms*—**ABC transporters, AlphaFold2, antifungal resistance, *Coccidioides immitis*, molecular dynamics, Valley fever.**

## I. Introduction

Valley fever, or coccidioidomycosis, is a fungal disease caused by *Coccidioides immitis* and *Coccidioides posadasii*, which are endemic to arid regions of the southwestern United States and parts of Central and South America. The disease is contracted through the inhalation of airborne fungal spores and can range from mild respiratory symptoms to life-threatening disseminated infections, particularly in immunocompromised individuals [1]. According to the CDC, approximately 20,000 Valley fever cases are reported annually in the United States, though the true number is likely much higher

due to widespread underdiagnosis [2]. In California alone, occurrences have dramatically increased from 2,300 cases in 2014 to over 12,600 cases in 2024 [3]. While standard antifungal therapies such as fluconazole and itraconazole remain widely used [4], emerging drug resistance in *Coccidioides* spp. poses a serious clinical threat. More than 37% of *Coccidioides* spp. clinical isolates exhibited increased fluconazole resistance under laboratory conditions [5], echoing resistance trends observed in other pathogenic fungi [6]. Despite the growing public health concern, however, little progress has been made involving studying resistance in this pathogen.

Fungal pathogens are an increasingly recognized global health threat, with antifungal resistance being a major clinical challenge [7]. However, much of the research to date has focused on a small number of model organisms, leaving many regionally significant fungi underexplored [8]. As cases continue to rise [2] and treatment options remain limited [1], there is an urgent need to systematically investigate *C. immitis* resistance mechanisms.

ATP-binding cassette (ABC) transporters, membrane-bound efflux pumps that expel antifungal compounds from the cell, are one of the key mechanisms of drug resistance in fungi. In well-studied species such as *Candida albicans*, ABC transporters like CDR1 have been directly linked to azole resistance [9]. The drug resistance-mediating transporters typically belong to the pleiotropic drug resistance (PDR) or the multidrug resistance (MDR) subfamilies [10]. However, no comprehensive structural characterization of ABC transporters exists for *C. immitis* [11], leaving a critical gap in the development of tailored treatments that can overcome drug resistance.

This study addresses this gap by proposing a structure-guided framework to assess the druggability of drug resistance-mediating ABC transporters in *C. immitis*, a clinically relevant but understudied pathogen. This paper makes two key contributions: the first structural analysis of *C. immitis* efflux pumps, releasing accurate AlphaFold2 models and 520 docked complexes, and an end-to-end, low-cost pipeline integrating pocket prediction, molecular docking, and 2 ns MD into an efficient early-stage pocket screening, filtering, and prioritization tool

for precision antifungal drug design.

## II. RELATED WORK

ABC transporters are key mediators of antifungal resistance across multiple pathogenic fungi. Peng *et al.* [9] used cryo-electron microscopy to show that CDR1, a major contributor to drug resistance in *C. albicans*, has a central drug-binding cavity with hydrophobic transmembrane residues that interact with azoles. The inhibitor milbemycin oxime bound this site, forming stabilizing interactions and blocking substrate entry and ATP hydrolysis. Ibe *et al.* [12] further highlighted CDR1's function, structure, and binding mechanisms, underscoring its role as a model for drug resistance-mediating ABC transporters. While extensively studied in *C. albicans*, efflux pumps in *C. immitis* are poorly characterized despite growing resistance, highlighting the need for structure-based investigation and inhibitor evaluation, such as with milbemycin.

Monroy *et al.* [11] uncovered a large number of transporter proteins in *C. immitis*, with 1,288 transporters categorized, including 44 putative ABC transporters identified. While sequence-based annotations may suggest functional similarities to known drug resistance-mediating transporters in other fungi, these proteins remain structurally uncharacterized. Their study underscores the need for further structural analysis of *C. immitis* transporters.

Donovan *et al.* [13] described the clinical spectrum of Valley fever, which ranges from mild symptoms to disseminated disease requiring long-term antifungal therapy. Approximately 30% of cases developed serious pulmonary complications while 10% progressed to severe forms of the disease with possibly life threatening consequences. Galgiani *et al.* [4] detailed current treatments of coccidioidomycosis and emphasized the difficulties in treating this fungal infection. Current therapies for more serious cases primarily involve common antifungal agents, such as triazole antifungals like fluconazole and itraconazole. Treatment with amphotericin B is reserved for more severe infections because of its adverse health effects. However, infections can persist or recur despite extended therapy, emphasizing the necessity of more effective and less toxic antifungal therapies tailored to patients.

These studies point out critical research gaps in the understanding of the pathogens responsible for Valley fever. *C. immitis* efflux pumps lack characterization despite plausible similarities to known drug resistance-mediating proteins like *C. albicans* CDR1. Current Valley fever treatments also suffer from limitations in both safety and efficacy, with reports of growing resistance only urging further action. This study addresses these gaps by providing the first structural characterization of *C. immitis* ABC transporters, combining structural modeling, druggable pocket prediction, and inhibitor testing. These efforts lay the groundwork for enhancing both current treatments and enabling the creation of precision health-centered therapies with AI-driven methods for drug development.

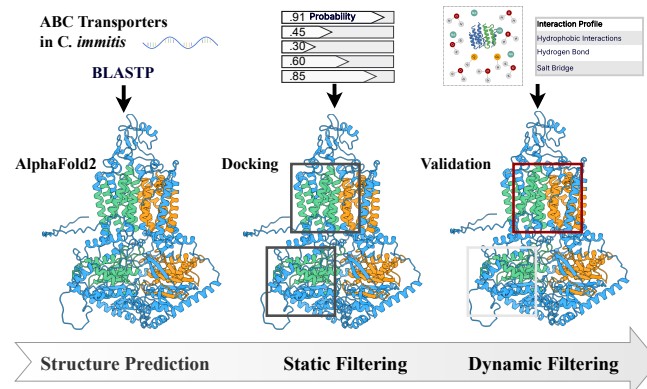

Fig. 1: Structure-guided computational framework for pocket screening with *C. immitis* ABC transporters: AlphaFold2 modeling, docking-based filtering, and short MD simulations for pocket validation.

## III. METHODS

To investigate antifungal resistance mechanisms in *C. immitis*, this study developed a multi-stage computational structure-guided framework. As illustrated in Fig. 1, the framework consists of three main stages: (1) protein selection and structure prediction, (2) static filtering through binding pocket prediction and molecular docking, and (3) dynamic filtering via short-timescale molecular dynamics (MD) simulations with interaction profiling. A retrospective CompositeScore analysis is presented. All curated datasets and related code are openly available at [14].

### A. Protein Selection and Characterization

ABC transporter candidates in *C. immitis* were identified using BLASTp searches against a curated panel of 23 antifungal resistance–mediating reference proteins. This panel included experimentally validated PDR-type and MDR-type efflux pumps from fungal species such as *C. albicans*, *Aspergillus fumigatus*, and *Trichophyton rubrum* [15]. Five ABC transporter candidates were selected based on their high sequence homology (65.7%–78.5%) to the compiled list of transporters.

Functional domain annotation was performed using Inter-Pro [16], confirming the presence of conserved ABC transporter features, including two nucleotide-binding domains (NBDs), two transmembrane domains (TMDs), Walker A motifs, and ABC signature motifs. The selected five transporters spanned both MDR and PDR subfamilies and exhibited conserved architectures, supporting their potential role in antifungal drug efflux.

The five ABC transporters from *C. immitis* were modeled using AlphaFold2-Multimer v3 via ColabFold v1.5.5 [17], and structural confidence was assessed using per-residue predicted local distance difference test (pLDDT) scores.

## B. Docking and Static Filtering

To probe inhibition across transporter subfamilies, five small molecules were selected based on prior evidence of their ability to inhibit fungal ABC transporters, chemical diversity, and safety for patients. Selection emphasized compounds with known activity against homologs identified in the BLASTp search.

Tacrolimus (FK506), an immunosuppressant, inhibits fungal transporters like CDR1 by impairing drug efflux and disrupting calcineurin signaling with azole synergy [18]. Verapamil, a calcium channel blocker and known ABCB1 (P-glycoprotein) inhibitor, synergizes with fluconazole while blocking efflux [19]. Beauvericin, a fungal mycotoxin, suppresses ABC transporter activity and boosts azole potency in resistant pathogens [20], [21]. Disulfiram, an anti-alcoholism drug, impairs ATP hydrolysis via conserved cysteines, raising intracellular antifungal levels [22]. Milbemycin oxime, a veterinary antiparasitic agent, binds the efflux channel and increases azole susceptibility in resistant isolates [9], [15], [23], [24]. Ligand dimensions were computed with RDKit [25] to confirm spatial coverage. Together, these five ligands provided chemical diversity and mechanisms well-suited for probing ABC transporter druggability.

Ligand-binding pockets were predicted using PrankWeb [26] and ranked by predicted binding probability and solvent-accessible surface (SAS) features. Grid dimensions for docking were dynamically scaled according to SAS point counts to account for pocket volume and geometry.

Molecular docking was conducted using AutoDock Vina [27], with standardized search parameters across all protein-ligand-pocket complexes and key parameters summarized in Table I. Binding poses were filtered using two criteria: docking affinity $\leq -8.0$ kcal/mol and pocket probability $\geq 0.20$ as predicted by PrankWeb.

TABLE I: Key Computational Parameters

| Module | Setting |
| --- | --- |
| Docking exhaustiveness (Vina) | 32 |
| Docking max. modes | 10 |
| MD timestep / length | 2 fs / 2 ns × 3 |
| Drift cutoff for stability | 2.5 Å |
| Force field | AMBER14SB / GAFF2.1 |
| Logistic regression ($C$ grid) | 5 (0.01–10) |

## C. MD Simulations and Validation

Shortlisted complexes underwent MD simulations (2 ns, triplicate) using OpenMM [28]. Each complex was solvated in a TIP3P water box with 0.15 M NaCl and simulated under NPT conditions at 300 K and 1 atm using Langevin dynamics and Particle Mesh Ewald (PME) electrostatics. Proteins were parameterized with the AMBER ff14SB force field and ligands were parameterized using GAFF 2.11 with AM1-BCC charges, and docked poses were used directly for MD system initialization.

Complexes were deemed stable with ligand center-of-mass drift < 2.5 Å and no significant loss of molecular contacts. In addition, Protein–Ligand Interaction Profiler (PLIP) [29] was used to monitor hydrogen bonds, hydrophobic contacts, and other key interactions.

Complexes were validated by aligning PDR-type transporters (CIMG_00533, CIMG_01418, CIMG_09093) to *C. albicans* CDR1 (PDB: 9IUM, milbemycin-bound) over the conserved domains, and MDR-type transporters (CIMG_00780, CIMG_06197) to human ABCB1 (PDB: 7OTG), a well-studied ABC drug efflux transporter. Alignments were assessed by comparing predicted pocket centers with the canonical substrate-binding cavities, and close overlays aided in retrieving proteins missed by static filtering or docking false negatives.

## IV. RESULTS AND DISCUSSION

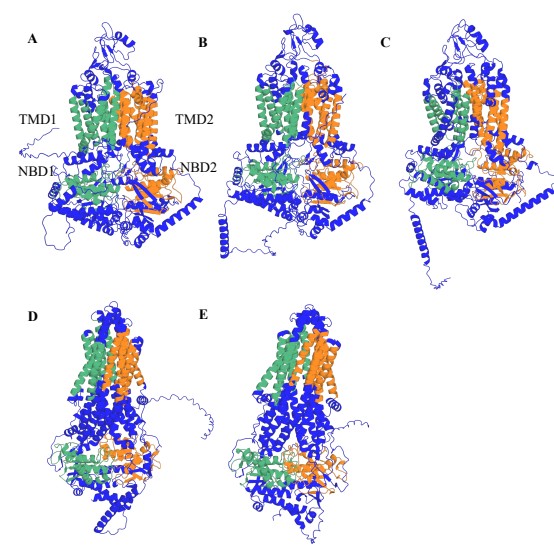

Fig. 2: Predicted 3D structures of ABC transporters in *C. immitis*. (A) CIMG_00533, (B) CIMG_01418, (C) CIMG_09093, (D) CIMG_06197, and (E) CIMG_00780.

A BLASTp search against the panel of antifungal resistance-mediating ABC transporters, including efflux pumps from clinically significant species like *C. albicans*, identified top-ranked homologs of *C. immitis*. The five selected candidates exhibited high sequence similarities to *T. rubrum* and *A. fumigatus*, as shown in Table II. The transporters were annotated using InterPro, confirming the presence of NBDs, TMDs, and conserved motifs.

Structural modeling produced transporter models with high structural confidence, with average per-residue pLDDT scores consistently exceeded 95. Protein models and conserved domains are shown in Fig. 2.

## A. Static Filtering via Docking and Pocket Scoring

This study evaluated 520 protein-ligand-pocket combinations across five ABC transporters, their predicted pockets, and five chosen ligands. While not as extensive as high-throughput efforts involving millions of compounds, this setup provides structural and dynamic context often absent from

TABLE II: Architecture and Confidence of ABC Transporters in *C. immitis*

| Gene | Protein Accession | Subfamily | Closest Homolog | NBD Domains | TMD Range | NBD pLDDT | TMD pLDDT |
|------|-------------------|-----------|-----------------|-------------|-----------|-----------|-----------|
| CIMG_00533 | XP_001246762.2 | PDR | *T. rubrum*/MDR1 (70.5%) | 172–330, 875–1026 | 514–1515 | 95.7 | 95.5 |
| CIMG_00780 | XP_001247009.1 | MDR | *T. rubrum*/MDR5 (75.0%) | 422–580, 1064–1215 | 80–1007 | 96.2 | 96.0 |
| CIMG_01418 | XP_001247647.1 | PDR | *T. rubrum*/MDR3 (78.4%) | 192–351, 887–1038 | 535–1498 | 95.8 | 95.9 |
| CIMG_06197 | XP_001242301.1 | MDR | *T. rubrum*/MDR2 (71.9%) | 447–604, 1116–1267 | 99–1056 | 96.1 | 95.9 |
| CIMG_09093 | XP_001239472.1 | PDR | *A. fumigatus*/ABCC (65.7%) | 141–300, 833–983 | 485–1450 | 95.7 | 95.6 |

brute-force docking workflows, enabling the screening and filtering binding sites beyond static scoring.

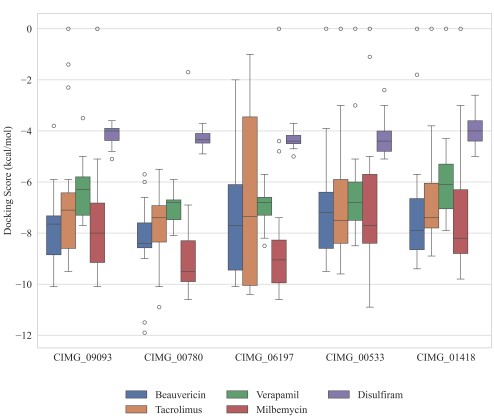

Fig. 3: Distribution of docking scores for the five ligand across all predicted pockets in each *C. immitis* ABC transporter. Lower scores indicate stronger predicted binding.

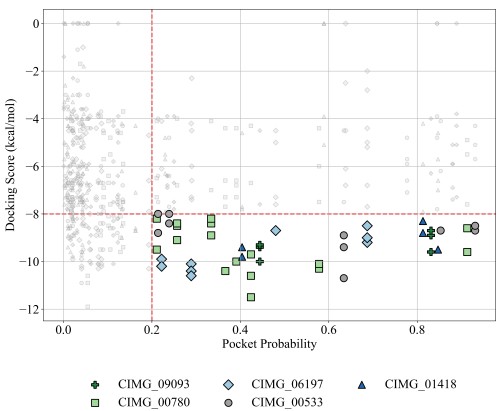

Fig. 4: Shortlisted protein–ligand–pocket combinations filtered by docking score $\leq -8.0$ kcal/mol and pocket probability $\geq 0.2$. Each point represents a docked complex, and combinations in the lower-right quadrant pass the filtering thresholds for downstream validation.

Docking boxes were dynamically configured using PrankWeb-derived pocket geometry. For each predicted pocket, PrankWeb outputs were used to define the grid center and quantify SAS points, providing geometry-aware coverage tailored to each pocket's volume and depth. To ensure adequate space for larger ligands, a minimum grid size of 28 Å was applied when pocket dimensions approached the size of the inhibitor, allowing room for ligand flexibility and pose variation. For smaller pockets and ligands, a general lower bound of 22 Å was enforced. These thresholds were applied automatically during batch input generation. Grid boxes ranged between 22-30 Å, with a substantial fraction exceeding the 28 Å threshold to accommodate larger ligands. Docking inputs were generated programmatically, enabling reproducible evaluation.

To shortlist ligand-pocket pairs for downstream validation, static filtering was applied using two criteria: a docking score $\leq -8.0$ kcal/mol and a pocket probability $\geq 0.2$. Fig. 3 shows the docking scores of each ligand for the five proteins. Milbemycin consistently yielded the lowest docking scores, suggesting stronger binding affinity, while disulfiram showed weaker scores across all transporters. Among the transporters, CIMG_00533 exhibited consistent, moderate docking scores for four out of five tested ligands, suggesting that it can accommodate a diverse range of compounds with similar affinity. Such uniformity may result from certain binding sites

that limit variation in binding poses or pocket architecture that lacks features contributing to ligand selectivity and specific interactions.

The docking threshold reflects a conservative cutoff typically associated with docking affinity in AutoDock Vina scoring, while the pocket probability threshold ensures structural plausibility and surface accessibility. A sensitivity analysis was performed to assess the impact of the docking score cutoff. Of the 520 evaluated complexes, 47% scored $\leq -7.0$ kcal/mol, 29% met the stricter $-8.0$ kcal/mol cutoff, and only 14% passed the highly conservative $-9.0$ kcal/mol threshold. Based on this distribution, $-8.0$ kcal/mol was selected as a balanced compromise, being stringent enough to exclude weak binders while retaining nearly one-third of the plausible candidates for MD validation, as shown in Fig. 4.

### B. Dynamic Filtering via Molecular Dynamics and Structural Mapping

To evaluate dynamic binding stability, 26 shortlisted protein-ligand complexes were subjected to 2 ns explicit-solvent MD simulations, each performed in triplicate. Complexes exhibiting ligand drift $< 2.5$ Å and no significant loss of molecular contacts were classified as stable. Table III summarizes the 2 ns MD-based dynamic classification of all evaluated combinations. A total of 17 pockets were identified as stable. Fig. 5 illustrates representative cases spanning stable,

## TABLE III: Dynamic Validation Results with Status Classification And Rationale

| Protein | Pocket | Pocket Prob. | Ligand | Affinity (kcal/mol) | Contacts | Drift (Å) | Δ Contacts | Status | Rationale |
|---|---|---|---|---|---|---|---|---|---|
| CIMG_00533 | 1 | 0.93 | Milbemycin | -8.7 | 3 | 2.47 | 2 | Stable | Marginal drift, contact gain |
| | 2 | 0.85 | Milbemycin | -8.7 | 6 | 1.89 | 0 | Stable | Low drift, contact gain |
| | 3† | 0.84 | Verapamil | -7.8 | 11 | 1.41 | -1 | Stable | Very low drift, retained pose, canonical substrate-binding cavity |
| | 5 | 0.63 | Milbemycin | -10.7 | 5 | 1.95 | -0.5 | Stable | Low drift, minor contact loss |
| | 6‡ | 0.24 | Milbemycin | -8.4 | 4 | 5.47 | -1.5 | Unstable | High drift > 5 Å, pose likely disrupted |
| | 7 | 0.21 | Milbemycin | -8.8 | 9 | 3.23 | 1.5 | Unstable | High drift > 3 Å |
| CIMG_00780 | 1† | 0.91 | Milbemycin | -9.6 | 7 | 2.38 | 2 | Stable | Low drift, contact gain, canonical substrate-binding cavity |
| | 2 | 0.59 | Milbemycin | -10.3 | 6 | 1.97 | 5 | Stable | Low drift, contact gain |
| | 3 | 0.42 | Beauvericin | -11.5 | 10 | 0.86 | -1 | Stable | Low drift, retained contacts |
| | 4 | 0.39 | Milbemycin | -10.0 | 8 | 12.44 | 2 | Unstable | High drift > 10 Å |
| | 5 | 0.37 | Milbemycin | -10.4 | 5 | 2.00 | 2 | Stable | Low drift, contact gain |
| | 6 | 0.33 | Milbemycin | -8.9 | 6 | 2.56 | 2 | Moderate | Drift > 2.5 Å |
| | 7 | 0.26 | Milbemycin | -9.1 | 4 | 14.72 | 3 | Unstable | Ligand displacement > 10 Å |
| | 8‡ | 0.21 | Milbemycin | -9.5 | 6 | 35.68 | -1.5 | Unstable | Ligand fully dissociated |
| CIMG_01418 | 1 | 0.85 | Milbemycin | -9.5 | 7 | 1.62 | 0 | Stable | Low drift, retained pose |
| | 2‡ | 0.81 | Milbemycin | -8.8 | 5 | 3.64 | -1 | Unstable | High drift > 3 Å, slight contact loss |
| | 3† | 0.59 | Verapamil | -5.1 | 7 | 1.58 | -0.5 | Stable | Low drift, slight contact loss, canonical substrate-binding cavity |
| | 4 | 0.40 | Milbemycin | -9.8 | 10 | 1.35 | 1 | Stable | Low drift, contact gain |
| CIMG_06197 | 1 | 0.69 | Milbemycin | -9.2 | 9 | 1.42 | 1 | Stable | Low drift, contact gain |
| | 2 | 0.69 | Milbemycin | -9.0 | 9 | 1.27 | -2 | Stable | Low drift, reduced contacts, canonical substrate-binding cavity |
| | 4† | 0.48 | Milbemycin | -8.7 | 10 | 1.91 | 3 | Stable | Low drift, contact increase, canonical substrate-binding cavity |
| | 6 | 0.29 | Beauvericin | -10.1 | 8 | 1.22 | -0.5 | Stable | Low drift, minimal contact loss |
| | 7‡ | 0.22 | Milbemycin | -10.2 | 6 | 2.44 | 0 | Stable | Marginal drift, contact retained |
| CIMG_09093 | 1† | 0.89 | Verapamil | -7.3 | 8 | 1.99 | -2 | Stable | Low affinity, moderate drift and contact loss, canonical substrate-binding cavity |
| | 3 | 0.83 | Milbemycin | -9.6 | 5 | 1.81 | -2 | Moderate | Low drift, contact loss |
| | 4‡ | 0.44 | Beauvericin | -10.0 | 12 | 1.05 | -3.5 | Unstable | Significant contact loss despite low drift |

† denote top-scoring candidate, shown as stable example in Fig. 5.

‡ Shown as unstable or marginal case in Fig. 5.

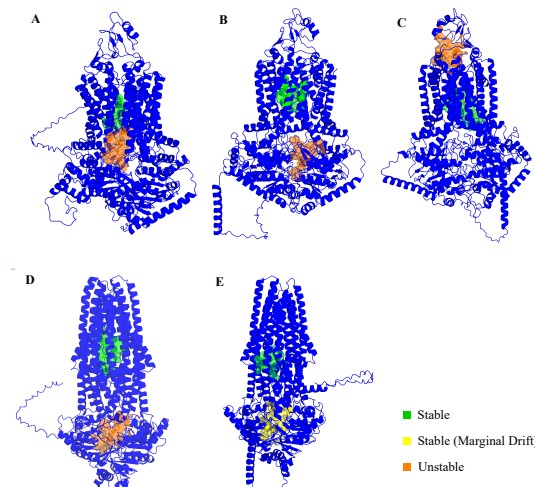

Stable
Stable (Marginal Drift)
Unstable

Fig. 5: Examples of predicted binding pockets in ABC transporters. (A) CIMG_00533, (B) CIMG_01418, (C) CIMG_09093, (D) CIMG_00780, and (E) CIMG_06197.

## TABLE IV: Extended 20 ns MD Validation of Representative Subset of Complexes

| Protein | Pocket | Ligand | 2 ns | 20 ns |
|---|---|---|---|---|
| *Initially stable at 2 ns*: remained stable at 20 ns | | | | |
| CIMG_00533 | 3 | Verapamil | 1.41 | 2.01 |
| CIMG_01418 | 1 | Milbemycin | 1.62 | 1.67 |
| CIMG_00780 | 3 | Beauvericin | 0.86 | 1.38 |
| CIMG_06197 | 2 | Milbemycin | 1.42 | 2.83 |
| *Moderate at 2 ns*: one remained, one drifted by 20 ns | | | | |
| CIMG_00780 | 6 | Milbemycin | 2.56 | 4.20 |
| CIMG_09093 | 3 | Milbemycin | 1.81 | 2.31 |

moderately stable, and unstable binding profiles. Milbemycin emerges as the most consistent inhibitor, exhibiting persistent interactions and low drift across multiple transporter sites. Its performance aligns with previously reported efflux-blocking activity against ABC transporters in other drug-resistant fungi, such as *C. albicans* and *T. rubrum* [9], [15].

To further assess whether 2 ns simulations are sufficient for early-stage screening, a representative subset of complexes (4 stable, 2 moderate) was extended to 20 ns unrestrained MD. This included Pocket 3 of CIMG_00533, which was initially

excluded for narrowly missing the docking score threshold (−7.8 kcal/mol vs. the −8.0 kcal/mol cutoff), but was re-included after structural alignment revealed close pocket centers, elaborated further in the following section. Fig. 6 shows drift stabilization during the first 2 ns of Verapamil in Pocket 3 of CIMG 00533. Table IV reports average drift over the full 20 ns simulations across three replicas for the transporter subset. The 4 stable complexes generally retained stability at the end of 20 ns MD. Pocket 2 of CIMG_06197 exhibited a drift of 2.83 Å, slightly exceeding the 2.5 Å early-stage cutoff but remaining below 3.0 Å, and thus can be considered marginally stable with the ligand largely retained in the pocket. In contrast, Pocket 6 of CIMG_00780, initially classified as moderate, exceeded 4 Å drift by 20 ns, indicating significant dissociation. These results reinforce 2 ns MD as an efficient filter to deprioritizes unstable pockets with early drift.

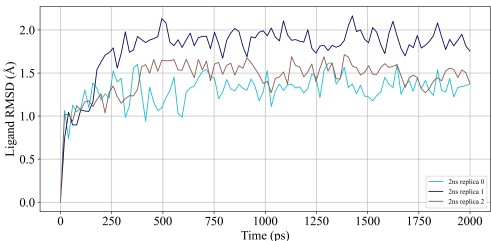

Fig. 6: Ligand drift trajectories from the first 2 ns of unrestrained MD of Verapamil in Pocket 3 of CIMG_00533. Each colored line denotes one replica; trajectories stabilize within ∼2 Å, illustrating the pocket's early binding persistence.

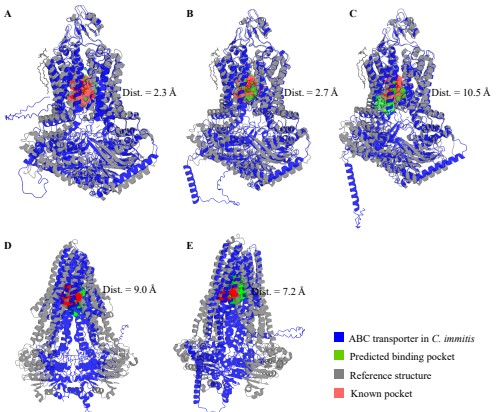

Fig. 7: Structural overlay of *C. immitis* ABC transporters with reference proteins CDR1 and ABCB1. (A) CIMG_00533, (B) CIMG_01418, (C) CIMG_09093, (D) CIMG_00780, and (E) CIMG_06197.

### C. Structural Alignment Confirms Pocket Similarity Across Fungal Transporters

Structure-based alignments against ligand-bound reference proteins validated pockets in *C. immitis* ABC transporters and recovered sites missed by static thresholds and docking false negatives, shown in Fig. 7. Measured distances between predicted pocket centers of CIMG_00533, CIMG_01418, and CIMG_09093 and the canonical substrate-binding cavity of CDR1 were 2.3 Å, 2.7 Å, and 10.5 Å, respectively. Pocket center distances between CIMG_00780 and CIMG_06197 and ABCB1 were 9.0 Å and 7.2 Å, respectively. While these distances indicate reasonable spatial correspondence, they also reflect limitations of cross-species alignment. Pocket 3 in CIMG_00533, initially filtered out due to moderate docking affinity, was recovered after close CDR1 alignment and confirmed by stable MD behavior, highlighting the value of multistep validation beyond rigid filtering.

### D. Stability Interpretation: CompositeScore

CompositeScore is a retrospective scoring function developed to interpret the outcomes of short MD simulations in terms of docking affinity, protein–ligand interaction patterns,

---

**Algorithm 1** Stability Interpretation

**Require:** Post-MD features: $\mathcal{T}$ (Vina), $\mathcal{I}$ (PLIP), $\mathcal{Q}$ (PrankWeb), $\mathcal{M}$ (MD drift)
1: **for** each pocket-ligand pair $i$ **do**
2:   $S_i \leftarrow 0.148\, z(-D_i) + 0.550\, z(P_i) + 0.297\, Q_i - 12.005\, z(\Delta_i)$
3: **end for**
4: **return** CompositeScore $\{S_i\}$ for retrospective ranking

---

pocket geometry, and observed ligand drift. Rather than serving as a predictive model, the score synthesizes simulation-derived features into a unified, interpretable metric. Its purpose is to reconstruct and explain which biophysical attributes contributed to the observed stability outcomes in retrospect. Algorithm 1 defines the scoring procedure.

*1) Score Definition and Label Assignment:* For each pocket–ligand pair $i$, the following features were used:
- $D_i$: AutoDock Vina binding affinity (kcal·mol$^{-1}$);
- $P_i$: PLIP total protein–ligand contact count;
- $Q_i$: PrankWeb pocket probability;
- $\Delta_i$: Ligand COM drift after 2 ns MD simulation.

Stability labels were derived from MD drift:

$$y_i = \mathbb{I}(\Delta_i \leq 2.5 \text{ Å})$$

CompositeScore was then defined as:

$$S_i = 0.148\, z(-D_i) + 0.550\, z(P_i) + 0.297\, Q_i - 12.005\, z(\Delta_i)$$

This formulation is diagnostic: $\Delta_i$ is included in the score but also used to define $y_i$, so predictive claims are avoided. The score is used only to interpret and rank simulation outcomes retrospectively.

*2) Validation and Feature Contribution:* When applied to the 26 MD-evaluated pocket–ligand pairs, CompositeScore achieved ROC-AUC of $0.86 \pm 0.03$ and retrospectively ranked all 17 stable poses above unstable ones. This performance reflects internal consistency, not predictive power. Since $\Delta_i$ is included both in score computation and label generation, CompositeScore cannot be used prospectively. Its purpose is diagnostic: (i) to quantify the contribution of each feature to observed stability, and (ii) to provide interpretable summaries for comparing transporter pockets and ligands post-MD. Coefficient magnitudes reflect PLIP interaction count is the dominant stabilizing term, while ligand drift strongly penalizes unstable poses. Pocket probability adds geometry-based context, complementing docking affinity.

Several limitations should be acknowledged to contextualize the findings and inform future development of antifungal therapies. This study focused on five ABC transporters and five ligands, which, while representative, do not capture the full diversity of resistance mechanisms or inhibitors. 2 ns trajectories are sufficient for finding early ligand–pocket dynamics but are unable to predict slower conformational changes involving interactions that may occur over extended timescales, limiting their ability to predict long-term pocket stability. Therefore,

this structure-guided framework is best suited for early-stage screening, where unstable pockets can be deprioritized while top-ranked candidates move to longer or enhanced testing. CompositeScore was designed as a retrospective tool to interpret MD outcomes and is not predictive. Addressing these limitations can further enhance future works and developments in precision antifungal drug design.

## V. Conclusion and Future Work

This study presents the first structure-guided characterization of ABC transporters in *C. immitis*, the causative agent of Valley fever, and provides a computational framework for investigating drug resistance–mediating ABC transporters. Five transporters were modeled, binding pockets were predicted, inhibitors were selected, and 520 protein–ligand–pocket combinations were screened. A shortlist of 26 complexes was validated through MD simulations, identifying 17 stable pockets in 2 ns screening across the transporters. Milbemycin emerged as the most consistently bound ligand, demonstrating stable binding as an efflux pump inhibitor. This work provides the first structural dataset for ABC transporters in *C. immitis* and integrates AlphaFold2 modeling, PrankWeb pocket prediction, AutoDock Vina docking, and short-timescale MD filtering into a unified screening workflow. The approach is optimized for early-stage pocket prioritization, which supports AI-integrated antifungal drug design efforts.

Future work will incorporate a broader range of ligands and resistance-associated transporters to enhance coverage. Machine learning-based early-stage stability prediction will be explored to streamline screening and reduce computational overhead. Together, these extensions aim to strengthen the structure-guided framework as an efficient screening tool within antifungal drug development pipelines, advancing precision treatment strategies.

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
