# OpenReview forum: "Structure-Guided Framework for Characterizing Drug Resistance-Mediating ABC Transporters in Coccidioides immitis"
_IEEE.org/EMBS/BHI/2025/Conference — BHI 2025_

### Official Review · Reviewer_i7Y9 · 2025-06-27

[review text omitted: it was posted to a different submission]

---

### Official Review · Reviewer_7CXh · 2025-07-12
**Structure-Guided Framework for Characterizing Drug Resistance-Mediating ABC Transporters in Coccidioides immitis**

**Confidence:** 2
**Clarity Of Writing:** fair
**Clinical Significance:** great
**Methodological Novelty:** good
**Overall Rating:** 4

**Experiments And Results:**

fair

**Questions For The Authors:**

I am wondering if there is any potential reason to see a different trend for CIMG_0053 docking score visualized in Fig.3, and if so, what are the plausible factors that may contribute to this observation?

**Strengths:**

The study provides a comprehensive introduction and literature review. Also, the study addressed an ongoing issue in the treatment of valley fever by proposing an approach for binding pocket assessment and inhibitor testing, as well as a new dataset, which can be seen as an impactful scientific contribution.

**Summary Of The Paper:**

The study provides a framework and dataset that contains the ABC and C. immits. The paper motivates the study by noting the sharp increase in drug resistance incidence in fungal disease. The study proposes a framework for binding pocket assessment and inhibitor testing and notes the dataset as a contribution.

**Weaknesses:**

One of my primary concerns is related to the potential limitations of this work, which has not been demonstrated comprehensively and needs careful consideration.

For a reader with a different background, the pocket assessment details provided in the abstract are not clear.

While the paper introduces a framework for binding pocket assessment and inhibitor testing, as well as a new dataset, the innovative and overall contribution of the paper in the conclusion section should be clarified based on the scope of the conference available on the conference webpage.

Also, the captions of the figures are not informative enough.

---

### Official Review · Reviewer_5MEP · 2025-07-15
**Structure-Guided ABC transporter profiling in Coccidioides immitis: Clinically impactful work although with limited methodological novelty**

**Confidence:** 4
**Clarity Of Writing:** good
**Clinical Significance:** great
**Methodological Novelty:** fair
**Overall Rating:** 6

**Experiments And Results:**

good

**Questions For The Authors:**

- Have you run any longer ($\geq 10 ns$) simulations on a subset of stable pockets to confirm that 2ns was sufficient? Results would strengthen confidence.

**Strengths:**

- The paper addresses a problem with high clinical relevance. It tackles the problem of drug resistance in valley fever pathogens, an area with significant public health impact.
- The paper produces actionable research outputs. It delivers specific transporter models, ranked pockets, and prioritized inhibitors that experimentalists can immediately test.

**Summary Of The Paper:**

The paper introduces a framework to probe how drug resistance works in the fungus Coccidioides immitis. The paper focuses on five ATP-binding cassette (ABC) transporters suspected of expelling therapeutics. The paper generates 3-D models, locates potential drug-binding pockets, and assesses a handful of known efflux-pump inhibitors in silico. By combining structure prediction, docking and molecular-dynamics, the paper identifies a set of transporter sites that appear most likely to block drug efflux.

**Weaknesses:**

- Limited methodological novelty: The paper primarily integrates established tools into a workflow without proposing new algorithms, modeling techniques, or scoring strategies. While the application is well-motivated and thorough, the contribution lies more in novel domain-specific execution than in methodological innovation.
- Short MD windows: Two-nanosecond simulations may miss slower rearrangements, stability conclusions could change at longer timescales.

---

### Official Review · Reviewer_apwf · 2025-07-18
**Review for Structure-Guided Framework for Characterizing Drug Resistance-Mediating ABC Transporters in Coccidioides immitis**

**Confidence:** 2
**Clarity Of Writing:** good
**Clinical Significance:** good
**Methodological Novelty:** good
**Overall Rating:** 7
**Final Rating:** 7

**Experiments And Results:**

good

**Questions For The Authors:**

None.

**Strengths:**

Clinical backgrounds and motivations are well-explained. Sufficient experiment results are provided.

**Summary Of The Paper:**

The paper presents a multi-stage structure-guided framework to investigate antifungal resistance mechanisms in C. immitis. The framework consists of three main parts: protein selection and structure prediction, static filtering, and dynamic filtering. The experiment results suggest that the proposed model can systematically analyze the binding abilities of ABC transporters.

**Weaknesses:**

None.